# Agricultural subsidies and global greenhouse gas emissions

David Laborde[1,2], Abdullah Mamun [1,2], Will Martin[1,2], Valeria Piñeiro[1,2] & Rob Vos [1,2✉]

Agricultural production is strongly affected by and a major contributor to climate change. Agriculture and land-use change account for a quarter of total global emissions of greenhouse gases (GHG). Agriculture receives around US$600 billion per year worldwide in government support. No rigorous quantification of the impact of this support on GHG emissions has been available. This article helps fill the void. Here, we find that, while over the years the government support has incentivized the development of high-emission farming systems, at present, the support only has a small impact in terms of inducing additional global GHG emissions from agricultural production; partly because support is not systematically biased towards high-emission products, and partly because support generated by trade protection reduces demand for some high-emission products by raising their consumer prices. Substantially reducing GHG emissions from agriculture while safeguarding food security requires a more comprehensive revamping of existing support to agriculture and food consumption.

[1] International Food Policy Research Institute (IFPRI), Washington, D.C., USA. [2] These authors contributed equally: David Laborde, Abdullah Mamun, Will Martin, Valeria Piñeiro, Rob Vos. ✉email: r.vos@cgiar.org

During 2017–2019, farm sectors in 54 major economies received together US$553 billion per year in the form of market price support and direct subsidies. Of this amount, US$446 billion (equivalent to 12.5% of gross farm receipts) was provided as direct subsidies from governments or as "market price support" that typically raises prices by restricting imports (Fig. 1 and OECD[1]). These direct subsidies are either "coupled" to output levels and input use, or "decoupled" from specific production and provided as direct payments to farmers. The 54 countries for which such data are collected by the OECD spent on average US$185 billion per year on coupled subsidies and US$68 billion per year on subsidies decoupled from production during 2017–2019. They further spent US$106 billion per year on General Services Support (GSS) policies designed to create enabling conditions for agriculture, such as agricultural innovation systems, sanitary and phytosanitary standards, and rural infrastructure.

The two components of support that influence output decisions most directly are subsidies coupled to output and market price support provided through trade measures. Coupled subsidies tend to increase output without lowering demand in the subsidizing countries and hence to increase global emissions. Market price support tends to increase supply in protecting regions but, at the same time, reduces demand for agricultural products in those countries by raising consumer prices, making its impact on global emissions an empirical question to be addressed in this paper. Decoupled support is designed to have no impact on output and, hence, unless accompanied by effective environmental conditions, also no effect on emissions. GSS support, however, includes investments in research and development (R&D) that may be reasonably assumed to reduce both the cost of production and emissions per unit of output.

A rough indicator of the relative magnitude of coupled price support and market price support is provided by dividing the value of producer support by the value of output at world prices, as shown in Table 1. A key feature of Table 1 is the extraordinary rate of border support in a few high-income countries, such as Japan (57%) and Norway (63%). Farm support rates in China are not as high, but nonetheless substantial as market price support

and coupled subsidies add almost 15% to farm output value. Also noteworthy is the negative market price support in India (−12% of farm output) combined with sizeable coupled subsidies (7%). Globally, the rate of support from coupled subsidies averaged 5.5% in 2017–2019, while market price support rates averaged 5.7%.

Meanwhile, greenhouse gas (GHG) emissions from agriculture are strongly concentrated in a few commodities with beef, dairy, and rice accounting for over 80% of agricultural GHG emissions (Table 2). The production of these emission-intensive goods is often heavily supported using market price-support measures. This suggests a clear link between agricultural support and GHG emissions. However, the strength of this link requires close examination. At least four factors need to be considered before making any strong inferences: (i) the average rate of support to agriculture, (ii) differences between types of support, (iii) differences in rates of support across commodities and countries, and (iv) impacts of support on production methods and processes.

The average rate of support to agriculture matters because high rates of support are likely to attract resources into agriculture, increase output and, at constant technology, increase emissions from production. The type of support matters because of its influence on overall incentives to both producers and consumers. Differences in rates of support across commodities may have important impacts on overall emissions given large differences in the emission intensity of commodities and across countries as measured by the $CO_2$ equivalent of greenhouse gases emitted per unit of output. As noted in a related study[2], output of individual

**Table 1 Coupled subsidies vs market price support, 2017–19 (support as % share of value of production at world market prices).**

|  | Coupled subsidies | Market price support |
|---|---|---|
| Australia | 1.4 | 0.0 |
| Brazil | 1.5 | 0.2 |
| Canada | 3.8 | 4.6 |
| China | 4.2 | 10.1 |
| EU28 | 9.8 | 4.1 |
| India | 6.7 | −12.1 |
| Japan | 8.6 | 57.0 |
| Mexico | 5.2 | 5.2 |
| Norway | 80.9 | 62.9 |
| Russia | 4.2 | 6.8 |
| South Africa | 1.1 | 3.4 |
| USA | 7.5 | 3.2 |
| Developed countries | 8.1 | 8.5 |
| Developing countries | 4.3 | 4.4 |
| Total | 5.5 | 5.7 |

Source: OECD[1].

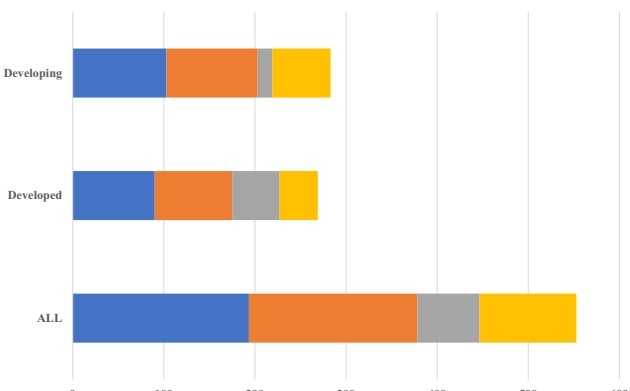

**Fig. 1 Agricultural producer support by main types of support, 2017–2019 (Values in billions of US$ per year).** The figure shows the total amount of annual government support to agriculture by type of support for 54 reporting countries. Support is presented for developed (high-income) and developing (low- and middle-income) countries by type: market price support ("MPS"), mainly consisting of border measures; subsidies coupled to input use or level of agricultural output ("coupled subsidies"); direct transfers to producers ("uncoupled subsidies"); and general service support expenditures ("GSSE"), which include other forms of support such as agricultural innovation systems, sanitary and phytosanitary standards, and rural infrastructure. Source: OECD[1].

**Table 2 GHG emissions (in $CO_2$ eq) from agriculture by commodity, 2014 (shares in percent).**

|  | Developed countries | Developing countries | World |
|---|---|---|---|
| Rice | 3.7 | 18.6 | 15.6 |
| Other cereals | 19.2 | 7.2 | 9.7 |
| Milk | 20.6 | 17.4 | 17.9 |
| Ruminant meat | 46.4 | 51.5 | 49.4 |
| Pig meat | 7.6 | 3.1 | 4.0 |
| Poultry meat | 1.3 | 1.3 | 1.3 |

Source: authors' calculations based on FAOSTAT.

agricultural commodities is likely to be more responsive to differentials in agricultural support rates than is overall agricultural output to the average rate of agricultural support. The same study also points out that, while there are many cases where high rates of support are paid on emission-intensive commodities, the average rate of support to high-emission commodities was below that for relatively low-emission-intensive commodities for almost all of the 1993–2015 period.

Support intended to influence production practices and processes, such as subsidies on fertilizers, pesticides, or improved seeds, also matter. In practice, these mostly aim to stimulate agricultural production which may induce more emissions unless improved practices are more resource efficient. Higher use of inputs such as fertilizer may be an additional source of GHG emissions though improved, climate-resilient seeds may help improve environmental outcomes. Some support programs, like the reformed Common Agricultural Policy of the EU, condition support on compliance with environment-friendly production processes and land conservation practices.

The OECD's GSS estimate includes support for activities such as agricultural research, development, and training intended to raise agricultural productivity. Productivity gains tend to reduce the emission intensity of agricultural production, for instance, through reduced use of intermediate inputs. Some new technologies appear able to both reduce emission intensities and lower costs (see, for example, Mernit[3]).

This article focuses on the implications of current agricultural support policies for GHG emissions. It applies a rigorous model-based analysis of the impacts of incentives on agricultural outputs and emissions. This analysis provides an opportunity to consider all the influences outlined above—impacts on overall output, differences in incentives across countries and commodities, as well as differences in farm technologies and practices used for production. It also allows us to examine the extent and potential implications of environmental conditionalities incorporated in producer support measures. We consider not just the total emissions per unit of output, but also the source of those emissions—whether they are, for instance, from enteric fermentation by ruminants or methane emissions from rice cultivation.

## Results

### The emission intensity of agricultural production

A key parameter for understanding the impacts of agricultural support on climate change is the emission intensity of production by region, measured by the amount (in kg) of $CO_2$-equivalent greenhouse gases produced per kg of output. If production of a good in an area with higher emission intensities is replaced by production from an area with lower-emission intensities, global emissions from that production of that commodity will fall for the same level of output. For a proper assessment of the quantitative impact on emissions of agricultural support measures, a general equilibrium approach of the type used in this paper is needed to account for possible shifts between products. If, for instance, a reduction in support for rice leads to an increase in beef production as resources are reallocated, global emissions may increase even if emissions from rice are reduced.

Emission intensities vary greatly across countries/regions and commodities (see Table 3 and the Methods section for further detail). The emission intensity for bovine meat is by far the largest for any food product, and it ranges from 12.1 kg $CO_2$ eq per kg of production in the United States to 108.3 kg $CO_2$ eq in India. There is a clear association between income levels and emission intensity, with the intensity for beef more than twice as high in the group of developing and emerging economies than in high-income, developed countries. Underlying this link is a strong relationship between productivity levels and emissions as productivity increases typically save on inputs and reduce emissions per unit of output. Tubiello[4] points out that total emissions from agriculture have fallen steadily since the 1980s in the countries subject to emission reduction commitments under the Kyoto protocol—despite substantial increases in incomes and population. Some progress on this front has been made in both developed and developing countries. While higher in most cases in developing than in developed countries, emission intensities have fallen much more rapidly in developing countries since the early 1990s[2].

### The impact of agricultural support on GHG emissions

We estimate the impact of current agricultural support measures on GHG emissions through simulations using IFPRI's global computable general equilibrium (CGE) model, MIRAGRODEP (see Methods section), augmented with models that capture the impacts of changes in outputs and inputs on emissions. We run simulations with the MIRAGRODEP model that compare observed levels of output and emissions by country and commodity with those that would come about in the absence of the government support. We look specifically at the impacts of coupled subsidies and border restrictions (trade measures) and simulate both the impacts of each type of support and their combined impacts. Supplementary Table 1 summarizes impacts on output, while Table 4 and Fig. 2 summarize the main results for emissions. More detailed findings are available in a related

**Table 3 Emission intensities for key products, countries, and country groupings, 2013–2015 (kg $CO_2$ eq. per kg of production).**

|  | Cereals excl. rice | Eggs | Bovine meat | Chicken | Pig meat | Milk | Rice |
|---|---|---|---|---|---|---|---|
| Australia | 0.3 | 0.4 | 20.2 | 0.2 | 2.5 | 0.7 | 0.7 |
| Brazil | 0.2 | 0.8 | 35.7 | 0.3 | 2.6 | 1.2 | 0.5 |
| China | 0.3 | 0.6 | 16.9 | 0.6 | 1.0 | 1.1 | 0.8 |
| EU | 0.2 | 0.7 | 15.4 | 0.3 | 1.6 | 0.6 | 3.0 |
| India | 0.3 | 0.5 | 108.3 | 0.5 | 5.0 | 1.1 | 0.7 |
| Indonesia | 0.2 | 1.0 | 42.8 | 3.6 | 4.9 | 2.9 | 1.1 |
| Japan | 0.2 | 0.4 | 9.5 | 0.3 | 0.9 | 0.3 | 0.8 |
| Mexico | 0.2 | 0.5 | 28.1 | 0.5 | 2.8 | 0.5 | 3.4 |
| Russia | 0.1 | 1.0 | 15.0 | 0.3 | 1.3 | 0.8 | 2.2 |
| USA | 0.2 | 0.5 | 12.1 | 0.3 | 2.0 | 0.4 | 1.1 |
| Developed countries | 0.2 | 0.6 | 15.3 | 0.3 | 1.7 | 0.6 | 1.2 |
| Developing countries | 0.2 | 0.7 | 32.2 | 0.7 | 1.4 | 1.3 | 0.9 |
| World | 0.2 | 0.7 | 26.5 | 0.6 | 1.5 | 1.0 | 0.9 |

Source: authors' calculations based on FAOSTAT. See Laborde et al.[5] for further detail.

**Table 4 Impact of current agricultural support (coupled subsidies and border measures) on GHG emissions from agriculture by source, 2014–16 (Kt of $CO_2$ eq.).**

| | All | Crop residues | Enteric fermentation | Manure | Rice | Synthetic fertilizer | Energy and other[a] |
|---|---|---|---|---|---|---|---|
| *Coupled subsidies* | | | | | | | |
| World | 34,420 | 2915 | 6016 | 3871 | 1041 | 10,138 | 10,439 |
| Developed | 18,116 | 1079 | 4107 | 2987 | 206 | 4942 | 4795 |
| Developing | 16,304 | 1836 | 1909 | 884 | 834 | 5197 | 5644 |
| *Border measures* | | | | | | | |
| World | −127,635 | −4129 | −91,043 | −39,624 | 1193 | −1203 | 7171 |
| Developed | −25,597 | −3115 | −11,644 | −9139 | −201 | −3042 | 1544 |
| Developing | −102,037 | −1013 | −79,399 | −30,486 | 1394 | 1839 | 5628 |
| *All support*[b] | | | | | | | |
| World | −102,071 | −1257 | −88,780 | −37,691 | 2331 | 7511 | 15,815 |
| Developed | −7590 | −1728 | −7529 | −6086 | 33 | 1811 | 5909 |
| Developing | −94,481 | 471 | −81,251 | −31,605 | 2298 | 5700 | 9906 |

Source: MIRAGRODEP simulations. See Laborde et al.[5] for further detail.
a"Energy & other", includes emissions from energy use, as well as from burning crops and Savanna.
b"All support" refers here to coupled subsidies and border measures. Please note that the columns do not add precisely because of nonlinearity in the relationships considered.

working paper[5]. Throughout the analysis, we make the standard economic assumption that changes in output prices result in movements along a supply curve with the underlying production technology remaining constant. We introduce innovations to the underlying technology separately, that is, as the outcome of investments in research and development—perhaps partly price-induced—that are directed towards solving problems such as increasing farm incomes, reducing consumer prices of food and/or reducing GHG emissions.

Our findings show that coupled subsidies stimulate agricultural output and emissions, while agricultural trade interventions reduce emissions (as compared with a situation without these interventions). Specifically, coupled subsidies increase global farm output volume by 0.9% (see Supplementary Table 1). Primarily because of this stimulus to production, GHG emissions from agriculture are 34,420 kt of $CO_2$ eq higher (an increase of 0.6%) than they would be without the coupled subsidies. The impact on emissions is smaller than the output effect, because the stimulus provided is less for the most emission-intensive products, such as beef and dairy products, and because the expansion of output resulting from subsidies is larger in richer countries with lower-emission intensities. Table 4 further shows that the impact of coupled subsidies on emissions is similar in magnitude for developed and developing countries. Close to a third of the increase results from stimulus to synthetic fertilizer use. The total impact of subsidies on emissions will be greater once emissions from land use change are added to these estimates.

In contrast, and perhaps surprisingly, current border measures have a minuscule impact on global output, raising it by a mere 0.1% globally, with output rising in developed countries by 0.6% and declining in developing countries by 0.1% (Supplementary Table 1). Border measures reduce emissions by 128 million tons or 2.1% (Table 4). The impact of these trade measures on global output is smaller than that of coupled subsidies, even though much more support is provided through protection. This is because protection raises consumer prices in the countries providing it, reducing demand in countries that protect agriculture, and hence the global demand for the affected commodities. Output in countries other than those providing protection is reduced by lower world prices. The impact on emissions is also influenced by shifts in the location of output. Border measures increase output in some high-income countries with relatively low-emission intensities while negative protection reduces output of high-emission-intensity bovine meat, as is the case in several developing countries.

Combined, coupled subsidies and border measures help increase global farm output by 1.1%, mainly driven by higher output in the developed countries. Differences in emission intensities and price-induced shifts in demand imply that current incentives reduce global GHG emissions by 102 million tons of $CO_2$ eq (1.7% of current levels) compared with a situation in the absence of such support: The support measures, on balance, provide incentives that shift production from relatively high emission-intensity countries, such as Brazil, to those with somewhat lower-emission intensity, especially high-income economies, such as the EU (Fig. 2). The highly concentrated nature of emissions by commodity and the large differences in emission intensities across countries plays a major role in this outcome. Substantially lower output of bovine meat in Brazil (18%), India (32%), and Australia (31%), only partially offset by higher output in the EU and China, determine most of the estimated impact of agricultural support measures on global emissions.

**The importance of efficiency improvements for reducing emissions**. In sum, while many have criticized current subsidy programs as contributing to global warming, our results suggest that simply abolishing current programs could, in fact, lead to slightly higher emissions. There are, however, many reforms that could be undertaken to improve the performance of agricultural support against goals such as improving economic efficiency, reducing poverty, and lowering emissions. One approach that would serve all three of these goals simultaneously might be to increase the support to agricultural R&D, and particularly R&D that focusses on reducing emission intensities. This might be done by repurposing some of the resources currently provided as distorting subsidies to R&D that is currently counted under the GSSE element of the OECD's measures of support. Because many studies indicate that the economic returns from R&D focused on increasing agricultural productivity are extraordinarily high[6], and agricultural productivity growth appears to have a much bigger impact on poverty reduction than productivity growth in other sectors[7], the required reallocation of resources might be relatively small.

Furthermore, there is every reason to believe that research focused on reducing agricultural emissions—or combinations of cost and emission reduction— would substantially reduce emission intensities. While research with a strong focus on emission reductions as well as productivity increases is relatively new, there are already promising new technologies and practices

### a. Major developed countries and country groupings

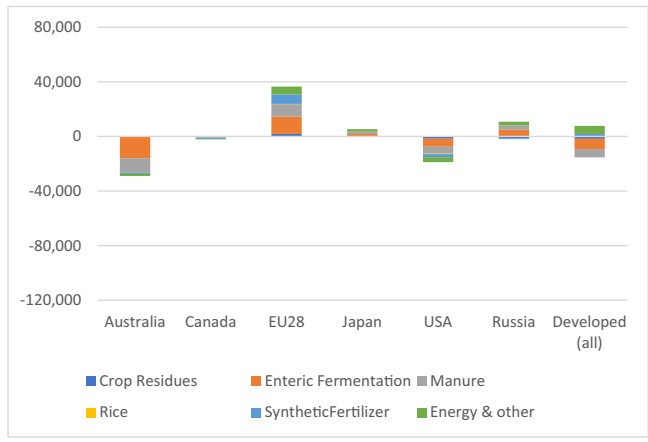

### b. Major developing and emerging economies

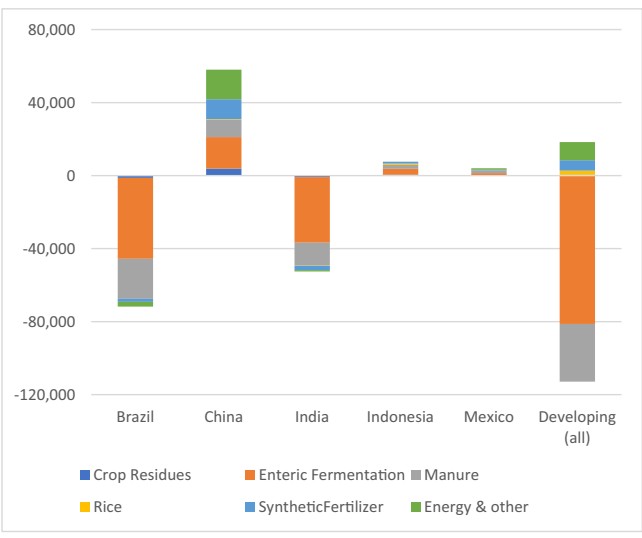

**Fig. 2 Impact of current coupled subsidies and border measures on GHG emissions by commodities and selected countries and country groupings.** The figure shows the results of model simulations measuring the combined impact of current support measures on greenhouse gas emissions from agricultural production. The impact is measured in kt of $CO_2$ eq by type of emission source (i.e., from crop residues, enteric fermentation by livestock, manure, methane emissions in rice cultivation, use of synthetic fertilizer, and use of energy and other sources). Impacts were estimated as the difference between actual emissions and a counterfactual scenario of what the level of emissions would be in the absence of support to agriculture in the form of coupled subsidies and border measures. Panel **a** shows the impacts for major developed countries and country groupings and panel **b** shows those impacts for major developing countries. Source: MIRAGRODEP simulations. See Laborde et al.[5] for further detail.

that could reduce in methane emissions from rice and from cattle by up to 50% (see, for example, refs. [3], [8] on dietary supplements for cattle and alternate wetting and drying in rice). Hurdles to adoption of some of these new technologies can be formidable[9], but many types of improved farm management practices could provide substantial environmental benefits at low cost. Because there has been relatively little emphasis in research programs on reducing GHG emissions, it seems likely that the portfolio of lower-emission innovations could be expanded quite rapidly if given greater priority. Innovations that reduce emissions from the

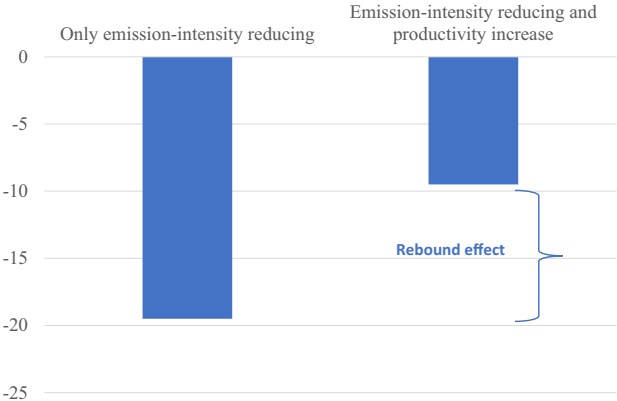

**Fig. 3 Impact of 30 percent reductions in emission intensities with and without agricultural productivity increases (percentage change from baseline).** The figure shows the impacts on global GHG emissions in two alterative scenarios. In the first scenarios ("only emission reducing") public support is redirected towards more R&D to achieve a 30% reduction in emission intensities and assuming new technologies become accessible at a low enough price to facilitate its widespread adoption. The second scenario ("emission intensity reducing and productivity increase") is the same as the first but adds that the assumption that new technologies would also reduce input needs, hence, reducing emission intensities while increasing productivity. Impacts for each scenario are measured in percentage change of GHG emissions with respect to the baseline. The "rebound effect" indicates that with the productivity increase the emission reduction will be lower, since higher productivity will allow lower agricultural prices and greater demand for produce, hence offsetting some the emission reduction achieved through technologies reducing emission intensities. Source: MIRAGRODEP simulations. See Laborde et al.[5] for further detail.

largest single source of GHG emissions—enteric fermentation by ruminants—would seem particularly likely to result in both emission reduction and increases in productivity since these emissions involve an obvious waste of a potentially valuable hydrocarbons.

As illustrative examples, we considered two polar cases: (i) R&D that reduces emission intensities by 30% in the countries for which we have data on agricultural incentives while reducing costs only by enough to permit its adoption. and (ii) innovations that reduce the need for all inputs, thus reducing emission intensities and increasing productivity. The first is perhaps like the case of dietary supplements for cattle, at least some of which reduce emissions without greatly stimulating output per unit of input. The second is more consistent with innovations like alternative wetting and drying in rice, which raises productivity of all inputs, while reducing emissions per unit. The 30% reduction is chosen because it seems to be within the range that is suggested as feasible by the currently available—but not yet widely adopted —innovations discussed above.

The results of these simulations are presented in Fig. 3. The reduction of almost 20% in global emissions from technical change that reduces emissions without raising productivity simply reflects our partial coverage of support measures. Because we account for production that accounts for roughly two-thirds of emissions, reductions in emission intensities of 30% on that production without any impact on productivity, result in a fall in global emissions of 20%.

The second innovation assumes a reduction is achieved in both emission-intensity and production costs by 30%. This would yield a reduction in global GHG emissions from agriculture of just under 10%. The fall by half in the emission reductions reflects the

rebound effect associated with innovations that reduce the costs of producing goods. Higher productivity reduces consumer prices and increases consumer demand. The increase in demand offsets some of the reduction in input use and in emissions induced directly by the technological innovation in this scenario. This estimate of the rebound effect from agricultural productivity is consistent with that obtained in another study using a quite different model to analyze the impacts of productivity growth in crops and livestock on greenhouse gas emissions[10]. Because price elasticities of demand for food are low, the rebound effect is small enough that a sizeable net reduction in emissions remains following the introduction of the innovation. This is in contrast with technical change in many other goods and services, such as the original case considered by Jevons[11], where improved steam engines lowered the cost of power from coal and increased demand enough to outweigh the reduction in the amount of coal needed for any given task.

The net reduction in emissions identified in this analysis should be considered a lower bound because it omits the gains from reductions in emissions from land-use change. With price inelastic demand for food, agricultural productivity improvements reduce the land footprint needed to meet food demand, adding a further reduction in emissions that we plan to evaluate in future work. Even with this important degree of underestimation of the net gains, both these experiments point to the potential for much larger reductions in emissions from productivity growth and, hence, from redirecting support measures to R&D and incentives to adopt climate-smart practices.

## Discussion

The analysis presented in this paper examines the implications of current levels of agricultural support on global GHG emissions from agricultural production. To assess these impacts, we compared the current level of emissions with a counterfactual without these support measures. In this assessment, we focused on emissions from agricultural production only. This allowed us to concentrate on the complexities associated with changing these subsidies and to provide a basis for understanding more comprehensive and far-reaching reforms. For the assessment we created a new database mapping GHG emissions by source, location, commodity, production stage and technology and incorporated this information into IFPRI's global model to relate agricultural production structures and market behavior to emission intensities by location, production sector, technology, and source of emissions.

Our findings show that current subsidies paid by governments that stimulate production induce both higher global agricultural output (0.9%) and emissions (0.6%). The existing market price support to farmers provided by trade barriers has almost no effect on global agricultural production and reduces GHG emissions by ~2% compared to a situation without such agricultural market protection. Combined, the coupled subsidies and border measures slightly increase global farm output (by 1.1%), while reducing global GHG emissions from agriculture by around 1.7%. These small net impacts arise because border measures in rich countries lower global demand more than they increase supply and induce shifts from relatively high emission-intensity producers to lower-emission-intensity producers in the rich countries. Coupled subsidies, by contrast, provide incentives to expand emission-intensive agricultural activities without any offsetting impact on demand. The upshot is that, on balance, current agricultural subsidies and trade protection appear to have a very modest impact on global emissions. This suggests that substantially reducing the vast current GHG emissions from agriculture will require an overhaul of current incentive structures,

shifting support to interventions that more directly target emission reduction, such as GHG taxes on output or consumer demand, or more funding for R&D in productivity-increasing and emission-savings technologies and subsidizing the cost of their adoption.

These findings are preliminary and further research is needed to understand the true impacts, especially since the present scenario analysis did not consider in the impacts on land use change or on the carbon sequestration capacity of forests and soils. Furthermore, the findings should not be taken as conclusions about the effectiveness (or lack thereof) of current agricultural support policies. Present agricultural support policies in most countries are largely based on political-economy considerations and rarely for their impacts on GHG emissions. Proper assessment of policy effectiveness requires assigning policies to the goal that they are to pursue most directly. Future work will also involve accounting for impacts of changes in support on land use change and the carbon sequestration capacity of forests and soils, as well as additional scenarios for repurposing subsidies in ways that are more sensitive to climate mitigation and adaptation of agricultural sectors. Our tentative conclusion is that simply abolishing current agricultural subsidies and market price support would at best have a very limited impact on emissions—and could even increase them slightly. This points to a need to investigate the use of multiple instruments focused on the multiple goals of policy makers—such as economic efficiency, emission reduction, food security, climate-resilience of production, and poverty reduction—if we are to successfully achieve these multiple goals.

## Methods

**Emissions database by drivers**. For this study, we created a new database of emissions in agricultural production. FAOSTAT presents vectors of data on emissions by type and by commodity for each country, but we need the full matrix of emissions by type, commodity, and source to allow us to consider changes in emissions by type in production of each commodity, such as reductions in emissions from enteric fermentation in beef production. Wherever possible, we derived this full matrix by reverse engineering the FAO emission data to ensure that the total matched the FAOSTAT estimates. Where this was not possible, as in the case of emissions from pesticides, we used a similar IPCC Tier 1 methodology to generate comparable estimates.

Emission sources are identified using eleven FAOSTAT-based categories included in Table 5 plus emissions from agricultural pesticides. The first step was to define the activity levels associated with commodity outputs, such as the area used for rice cultivation. The second was to calculate the emission coefficients (EC) for $CH_4$, $CO_2$, and $N_2O$ by activity level using, wherever possible, the FAOSTAT database. Finally, emissions of $N_2O$ and $CH_4$ were converted to $CO_2$ equivalents using 310 and 21 for $N_2O$ and $CH_4$ respectively.

In many cases, the FAOSTAT emission database provided implied emission factors by activity and emission source, such as the area harvested in rice cultivation and the nitrogen content of manure. In some cases, it provides the base activity data, such as areas of organic soil cultivation, and the number of head of livestock for enteric fermentation and manure management. In other cases, such as burning crop residues, only data on biomass burned are provided, rather than data on the crops burned. In such cases, we imported base activity data from the FAOSTAT crop and livestock production database for the crops whose residues are frequently burned—maize, rice, sugar cane, and wheat.

For synthetic nitrogen fertilizer, the activity data (i.e., agricultural use of nitrogen) is missing. We obtained fertilizer use data from two sources – FAOSTAT (http://www.fao.org/faostat/en/#home) and the International Fertilizer Association (www.ifastat.org). FAOSTAT gives the total fertilizer volume for many countries, while the IFA's Fertilizer Use by Crop data provide the nutrient content of fertilizer by crop for 54 countries. Fertilizer use data from FAOSTAT were scaled to match IFA numbers for all countries and this was done by mapping the characteristics of IFA countries to the countries listed in FAOSTAT. Finally, we estimated emissions by multiplying fertilizer volume by the emission coefficients given in FAOSTAT database. For the final version of the database, we retained the base activity (or index) data to estimate the average amount of emissions per index type (land, animals, output, fertilizer and energy). The process for creating this new database is presented schematically in Fig. 4.

To allocate emissions from enteric fermentation and manure management between the joint products of meat and milk (and wool in the case of sheep) from buffaloes, camels, cattle, goats, and sheep in line with the value of their products. The resulting livestock numbers were then linked to emissions using data from the

**Table 5 Shares of GHG emissions from agriculture by commodity and source, 2015 (% of total, excluding energy).**

|  | Rice | Other cereals | Milk | Ruminant meat | Pig meat | Poultry meat | Eggs | Total |
|---|---|---|---|---|---|---|---|---|
| Burning crops | 0.2 | 0.5 | 0.0 | 0.0 | 0.0 | 0.0 | 0.0 | 0.7 |
| Crop residue | 1.3 | 3.1 | 0.0 | 0.0 | 0.0 | 0.0 | 0.0 | 4.4 |
| Enteric fermentation | 0.0 | 0.0 | 11.0 | 30.5 | 0.6 | 0.0 | 0.0 | 42.1 |
| Manure management | 0.0 | 0.0 | 1.6 | 2.4 | 2.8 | 0.4 | 0.3 | 7.5 |
| Manure left on pasture | 0.0 | 0.0 | 3.6 | 13.3 | 0.0 | 0.7 | 0.4 | 18.0 |
| Manure applied to soils | 0.0 | 0.0 | 1.0 | 1.1 | 0.9 | 0.7 | 0.4 | 4.2 |
| Pesticides | 0.2 | 0.8 | 0.0 | 0.1 | 0.0 | 0.0 | 0.0 | 1.1 |
| Rice cultivation | 12.6 | 0.0 | 0.0 | 0.0 | 0.0 | 0.0 | 0.0 | 12.6 |
| Synthetic fertilizers | 2.4 | 6.5 | 0.0 | 0.7 | 0.0 | 0.0 | 0.0 | 9.6 |
| Total | 16.6 | 10.9 | 17.1 | 48.1 | 4.3 | 1.8 | 1.1 | 100.0 |

Source: authors' computation. Note: data in the table are global averages.

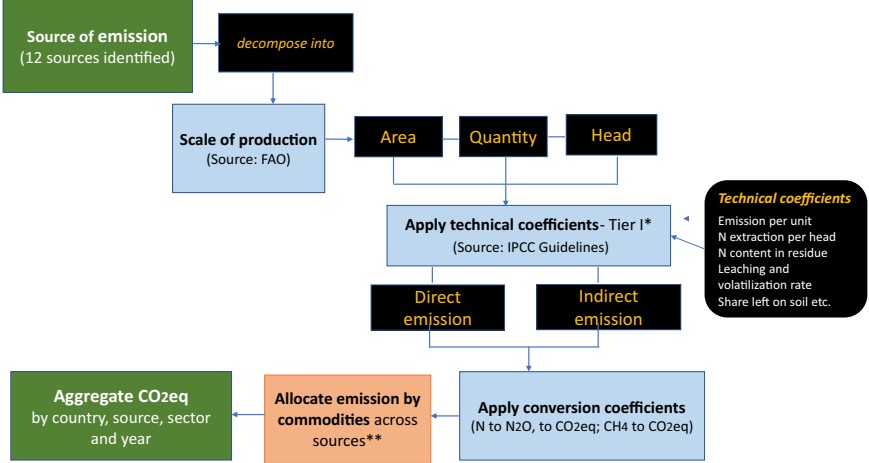

**Fig. 4 Creation of database of GHG emission from agriculture.** The figure shows the various steps in creating the database of GHG emissions from agriculture by source, location, commodity, production stage, and technology. Tier I (*) refers to default emission factors as defined in the 2006 guidelines of the Inter-governmental Panel for Climate Change (IPCC)[17]. The step allocating emissions by commodities and by source of emissions (**) involved using a matrix of disaggregated space and linkage. Source: authors' depiction.

FAOSTAT emissions database. In the final step we produced emissions data by country, emission source, and commodity. A summary of the overall structure of the emission shares is presented in Table 5.

**Modeling approach**. To assess the impacts of current agricultural support, we examine the implications of moving from current support levels to a hypothetical situation in the absence of intervention. For this analysis, we use IFPRI's global computable general equilibrium (CGE) model, MIRAGRODEP. It is an extension of the widely used MIRAGE model of the global economy[12]. The model was developed and improved with the support of the African Growth and Development Policy Modeling Consortium (AGRODEP). It is a multi-region, multi-sector, dynamically recursive CGE model. The model allows for a detailed and consistent representation of the economic and trade relations between countries[13]. In each country, a representative consumer maximizes a Constant Elasticity of Substitution-Linear Expenditure System (CES-LES) utility function subject to an endogenous budget constraint to generate the allocation of expenditures across goods. This functional form replaces the Cobb-Douglas structure of the Stone-Geary function (that is, LES) with a CES structure that retains the ability of the LES system to incorporate different income elasticities of demand[14], with those for food typically lower than those for manufactured goods and services. The demand system is calibrated on the income and price elasticities estimated by Muhammad et al.[15]. Once total consumption of each good has been determined, the origin of the goods consumed is determined by another CES nested structure, following the Armington assumption of imperfect substitutability between imported and domestic products.

On the production side, demands for intermediate goods are determined through a Leontief production function that specifies intermediate input demands in fixed proportions to output. Total value added is determined through a CES function of unskilled labor and a composite factor of skilled labor and capital. This specification assumes a lower degree of substitutability between the last two production factors. In agriculture and mining, production also depends on land and natural resources.

The underlying database used for the analysis is Pre-release 3 of the GTAP v10 database for 2014 (www.gtap.org). This database includes 141 regions/countries and 65 products. It includes updated Social Accounting Matrices for all individually specified countries and updated estimates of agricultural support measures based on measures of average domestic support provided by OECD[1], but adjusted to include the impacts on bilateral protection rates of major trade preferences. A realistic baseline is constructed aligned with the United Nations' demographic projections and updated IMF economic growth estimates to bring the base year values (2011) to those of the actual year of simulation (2020)

The data on agricultural support were adjusted in line with the measures discussed in the article for agricultural border measures and subsidies that influence output or input decisions (coupled subsidies). The model was augmented with a post-solution module based on the new emission database presented above and which links GHG emissions to output and inputs of agricultural activities determined in the model. This linkage is presented schematically in Fig. 5. The combined model was then used to assess the impacts of policy reform on emissions of $CH_4$, $CO_2$, and $N_2O$, and these results combined to generate a total $CO_2$ equivalent.

The macroeconomic assumptions used for the analysis were designed to be relatively "neutral" to avoid situations where macroeconomic adjustments such as real exchange rate changes outweigh the impacts of interest, and to allow us to focus on the impacts of agricultural support policies on emissions. These assumptions were:

(i)   no dynamic effects of investment decisions (the static version of the model was used);
(ii)  aggregate real public expenditures are kept constant and a consumption tax is adjusted to keep the government budget balance fixed as a share of GDP;
(iii) land use is constant to focus on emissions from agricultural production; and
(iv)  total employment is constant.

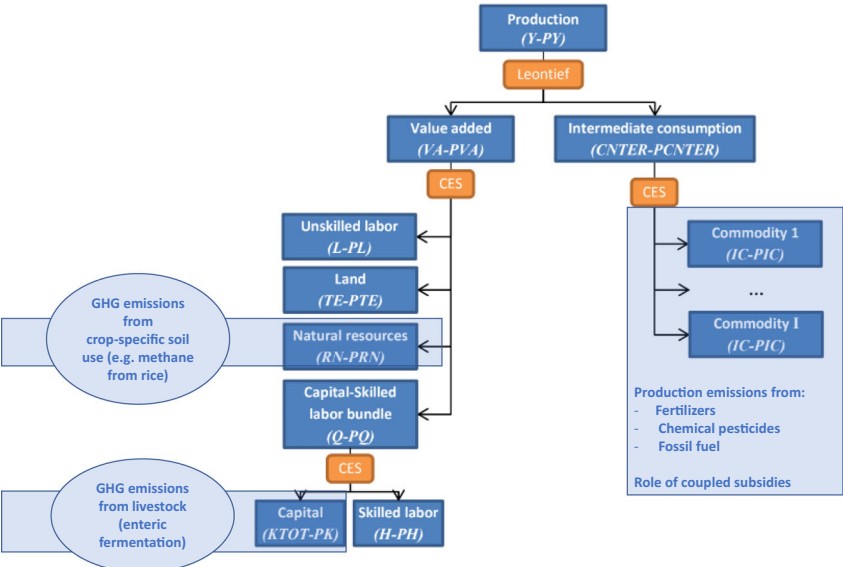

**Fig. 5 Linking the emissions module to the production system as captured in IFPRI's MIRAGRODEP model.** The flow chart shows the key production and natural resource modules of IFPRI's MIRAGRODEP model in the dark blue boxes and these are linked to the database of GHG emissions by commodity, source of emissions, and country (light blue box and ellipses). Source: authors' depiction.

Our approach of holding land use constant is consistent with many other studies in this area (e.g., refs. [16], [17]) and allows us to focus on changes in emissions from agricultural production, without needing to address the impacts of land use change, which are very context specific. Having estimates of the impacts on agricultural emissions is an important building block towards a full understanding of the impacts of reform. In this paper, we begin by considering the impact of removing coupled subsidies, and then turn to border measures.

## Data availability

Extended data and supplementary information related to this article are documented in the following papers available at https://www.ifpri.org/publication/reforming-agricultural-subsidies-improved-environmental-outcomes and https://doi.org/10.2499/p15738coll2.133852.

Source data for the newly created emissions database that supports the analysis of this study is publicly available in IFPRI's datasets repository, available at https://doi.org/10.7910/DVN/81RZBS (for the data) and https://doi.org/10.2499/p15738coll2.134270 (for the documentation).

## Code availability

GAMS (27.0.2) was used to run the MIRAGRODEP model, in combination with the CONOPT4 solver. The computer code (in GAMS) of the version of the MIRAGRODEP model is available at: http://www.agrodep.org/model/miragrodep-model

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

## Acknowledgements

This article was written as part of work undertaken for the Food and Land Use Coalition and a World Bank project on the "Environmental Impacts of Agricultural Support: Aligning Food Security and Climate Protection Objectives." Funding for this research was provided by The World Bank and the CGIAR Research Program on Policies, Institutions and Markets (PIM). The authors are grateful to the editor of Nature Communications, two anonymous referees, Johan Swinnen, as well as Madhur Gautam, Raffaello Cervigni, Richard Damania, Mike Toman, Stephen Ling, Dina Umali-Deininger, and Sergiy Zorya of the World Bank and to participants at an IFPRI policy seminar (21 April 2020); an NBER conference (30 April 2020); and the GTAP Conference (17 June 2020) for helpful comments on earlier versions of this study.

## Author contributions

All authors. D.L., A.M., W.M., V.P. and R.V. have contributed substantially and equally to the manuscript in terms of concept development, scenario design, analysis of model simulation results, and final write-up.

## Competing interests

The authors declare no competing interests.

**Additional information**

