## [Peer Review File · Nature Communications]

REVIEWER COMMENTS

Reviewer #1 (Remarks to the Author):

This paper supplies an overview about the effects of agricultural subsidies on global greenhouse gas emissions by concerning on four major factors, the average rate of support to agriculture, differences between types of support, differences in rates of support across commodities, and impacts of support on production methods and processes. This study enriches our knowledge about the effects of agricultural subsidies, but I think its marginal contribution is not specific and not efficient on its current vision.

Reviewer #2 (Remarks to the Author):

Referee Report

Agricultural subsidies and global greenhouse gas emissions

Summary:

This paper looks into the link between agricultural support (various subsidies) and the greenhouse gas (GHG) emissions. The author(s) argue that considering the large amounts of subsidies paid worldwide to agricultural sector there is a logical link with the GHG emissions from agriculture and that there is a lack of empirical evidence on the quantification of the link. Thus, the paper's contribution is in providing novel empirical evidence on the GHG emissions from agriculture that are due to subsidies. The paper does offer interesting results and discussion on an important environmental issue.

Main comments:

1. The paper offers a useful contribution to an important and policy relevant issue. The finding that agricultural subsidies mitigate GHG emissions to some extent is important and perhaps surprising at first sight. However, theoretically it is only logical to expect that well targeted agricultural subsidies aimed at dealing with market imperfections are bound to improve productivity and thus reduce GHG emissions. Certainly, this inference is specific to agricultural sector and food demand being relatively less elastic compared to demand for other goods.
2. Thus, the contribution of the paper as argued by the author(s) is mainly empirical. To the best of my knowledge, this is a novel study in utilising relevant data in an innovative way to estimate aggregate effects within a general equilibrium framework.
3. A weak point of the paper is presentation. While the topic of the paper is of high interest and the analysis is generally relevant and as far as I can tell well executed, presentation is not ideal. There are gaps that make the paper not very clear, especially in terms of missing detail in various key stages. Referring the reader for details to other publications is not a good practice in my view.

Minor comments:

1. Author(s) need to clarify why the focus of the empirical analysis is only on coupled subsidies and market price support. There are also decoupled subsidies which are increasing in importance especially in the EU.
2. When analysing the impact of subsidies, the assumption of constant technology is questionable. One of the main effects of subsidies on productivity is through technological change.
3. Often in the paper the term 'incentives' is used apparently as an alternative to the term 'subsidies'. Considering that out of all sorts of subsidy the empirical analysis is based on coupled subsidies, a more precise formulations should be used.
4. In the theoretically inclined discussion in section "Emission intensity of agricultural production" often the argumentation seems to rely on a partial equilibrium logic. For example, it is not certain that if agricultural activity with high emission at certain region is shut down the net emissions in that region will be reduced; the production resources could be relocated to another production process which may not be less emitting. The discussion should reflect more closely the general equilibrium empirical framework.

5. The discussion on channels and sources of improvement/reduction in emissions through improvements in productivity is not very systematic and clear. Even though authors present examples the theory behind the effects should be presented more clearly/explicitly.

6. Land use adjustments could be important for the scenario of no subsidies considering that subsidies induce changes in input demand and land is an important input in agricultural production. Author(s) at least should acknowledge that the emission effects are static or partial rather than net.

REVIEWER COMMENTS

Reviewer #1 (Remarks to the Author):

This paper supplies an overview about the effects of agricultural subsidies on global greenhouse gas emissions by concerning on four major factors, the average rate of support to agriculture, differences between types of support, differences in rates of support across commodities, and impacts of support on production methods and processes. This study enriches our knowledge about the effects of agricultural subsidies, but I think its marginal contribution is not specific and not efficient on its current vision.

Reviewer #2 (Remarks to the Author):

Referee Report

Agricultural subsidies and global greenhouse gas emissions

Summary:

This paper looks into the link between agricultural support (various subsidies) and the greenhouse gas (GHG) emissions. The author(s) argue that considering the large amounts of subsidies paid worldwide to agricultural sector there is a logical link with the GHG emissions from agriculture and that there is a lack of empirical evidence on the quantification of the link. Thus, the paper's contribution is in providing novel empirical evidence on the GHG emissions from agriculture that are due to subsidies. The paper does offer interesting results and discussion on an important environmental issue.

Main comments:

1. The paper offers a useful contribution to an important and policy relevant issue. The finding that agricultural subsidies mitigate GHG emissions to some extent is important and perhaps surprising at first sight. However, theoretically it is only logical to expect that well targeted agricultural subsidies aimed at dealing with market imperfections are bound to improve productivity and thus reduce GHG emissions. Certainly, this inference is specific to agricultural sector and food demand being relatively less elastic compared to demand for other goods.

RESPONSE: Thank you for this important point, which we now highlight this more explicitly in the section on efficiency improvements. We agree that the rebound effect is smaller in agriculture than for other goods, such as energy, because of the relatively low elasticities of demand for food. In addition, we now make the point that the estimates from agricultural production presented in this paper are a lower bound for the total gain associated with agricultural productivity growth because they omit gains from reductions in land use change. Precisely because, as you emphasized, the elasticities of demand for food are low, improvements in agricultural productivity reduce the amount of land needed to meet food demand and hence reduce the emissions from land use change.

2. Thus, the contribution of the paper as argued by the author(s) is mainly empirical. To the best of my knowledge, this is a novel study in utilising relevant data in an innovative way to estimate aggregate effects within a general equilibrium framework.

RESPONSE; Thank you very much for this very helpful observation.

3. A weak point of the paper is presentation. While the topic of the paper is of high interest and the analysis is generally relevant and as far as I can tell well executed, presentation is not ideal. There are gaps that make the paper not very clear, especially in terms of missing detail in various key stages. Referring the reader for details to other publications is not a good practice in my view.

RESPONSE: Thank you for providing these comments on the presentation. We have adjusted relevant parts of the presentation to avoid the reader having to go to other papers in order to follow the full argument of the paper itself.

Minor comments:

1. Author(s) need to clarify why the focus of the empirical analysis is only on coupled subsidies and market price support. There are also decoupled subsidies which are increasing in importance especially in the EU.

RESPONSE: We do include these measures in our estimates of total support in Figure 1. However, we now note that, because they are decoupled from output, they do not affect the output levels that are the main drivers of changes in emissions in our modeling framework. We now also clarify that the General Support estimates that we present include the research and development that we consider later in the paper.

2. When analysing the impact of subsidies, the assumption of constant technology is questionable. One of the main effects of subsidies on productivity is through technological change.

RESPONSE: In direct response to your comment, we clarify that, in our simulations, we are making the standard economic assumption that changes in output prices result in movements along supply curves with the underlying technology remaining constant. Because we do not have a reliable model of the process by which innovations are created, we prefer to introduce innovations in technology separately, that is, as the outcome of investments in research and development that are perhaps partly price induced. We examine the impacts of those outcomes explicitly, finding extremely interesting results with much lower “rebound effects” than often observed for R&D effects in other sectors of economic activity.

3. Often in the paper the term ‘incentives’ is used apparently as an alternative to the term ‘subsidies’. Considering that out of all sorts of subsidy the empirical analysis is based on coupled subsidies, a more precise formulations should be used.

RESPONSE: A central distinction in the paper is that between subsidies paid by treasuries—such as an output subsidy of \$100 per tonne of output— and market price support provided by mechanisms such as an import duty. We now more clearly make this distinction in the abstract and maintain it throughout the paper. This distinction is important because market price support both increases supply in the affected countries and reduces demand. The reduction in demand is not present in the case of subsidies, which is why subsidies contribute more to emissions than market price support.

We do occasionally use the more general term “incentives” because it encompasses a broader range of situations, such as where domestic prices are depressed by negative market price “support” resulting from export restrictions. The “incentives” term is also useful when a change creates impacts

in more than one direction, as when Market Price Support increases output but reduces consumer demand.

We now discuss decoupled subsidies in more detail than previously. They still do not feature in the simulations because they are explicitly designed with the intention of avoiding any impact on output.

4. In the theoretically inclined discussion in section “Emission intensity of agricultural production” often the argumentation seems to rely on a partial equilibrium logic. For example, it is not certain that if agricultural activity with high emission at certain region is shut down the net emissions in that region will be reduced; the production resources could be relocated to another production process which may not be less emitting. The discussion should reflect more closely the general equilibrium empirical framework.

RESPONSE: Thank you for highlighting the importance of general equilibrium analysis, and for the suggestion to strengthen this section, which we have done accordingly.

5. The discussion on channels and sources of improvement/reduction in emissions through improvements in productivity is not very systematic and clear. Even though authors present examples the theory behind the effects should be presented more clearly/explicitly.

RESPONSE: Thank you for pointing to the need to extend and clarify this section, which we probably wrote too concisely in the first version of the paper. In the new version of this section, we now spell out in greater detail the nature of the “rebound” problem associated with innovations that increase productivity and lower consumer prices. We also use the excellent point you made in your major comment 1, that the rebound effect is smaller for food than for other commodities because of the unusually low elasticities of demand for food.

We also point out that our estimate of the reduction in emissions is almost certainly a lower bound estimate. Once land use change is introduced—a major step we plan we plan to take in future work—the low elasticity of demand for food means that total land use in agriculture will decline, adding a reduction in emissions from land-use change to the already substantial reductions in emissions from agricultural production.

6. Land use adjustments could be important for the scenario of no subsidies considering that subsidies induce changes in input demand and land is an important input in agricultural production. Author(s) at least should acknowledge that the emission effects are static or partial rather than net.

RESPONSE: Thanks for this very important point. We think that this will be quantitatively most important in the case of productivity growth and we now acknowledge this point very thoroughly in that section. While we can’t put a specific number on this effect, there is no ambiguity about its direction, such that we are able to show that our current estimate of the effect is a lower-bound estimate, and, hence, we recommend its use as such for policy decision-making.